# Antibiotic Therapy and Athletes: Is the Mitochondrial Dysfunction the Real Achilles’ Heel?

**DOI:** 10.3390/sports10090131

**Published:** 2022-08-31

**Authors:** Valentina Puccini

**Affiliations:** Department of Clinical and Experimental Medicine, University of Pisa, 56126 Pisa, Italy; valentinapuccini20@gmail.com

**Keywords:** antibiotics, mitochondria, reactive oxygen species, skeletal muscle, tendon

## Abstract

It is widely recognized that athletes consume oral antibiotics almost twice as often as observed in the non-sports population in order to reduce as much as possible the period of inactivity due to bacterial diseases. However, increasing evidences have demonstrated the ability of some classes of antibiotics to induce muscle weakness, pain, and a feeling of fatigue upon resuming physical activity conditions that considerably limit the athletic performance of athletes, ascribable to alterations in the biochemical mechanisms underlying normal musculoskeletal activity, such as mitochondrial respiration. For this reason, tailoring a treatment plan for effective antibiotics that limit an athlete’s risk is paramount to their safety and ability to maintain adequate athletic performance. The present review illustrates and critically analyzes the evidence on the use of antibiotics in sports, deepening the molecular mechanisms underlying the onset and development of muscle–tendon alterations in athletes as well as delineating the pharmacological strategies aimed at counteracting such adverse events.

## 1. Introduction

Despite more athletes being generally young and healthy and thus are thought to ingest few or no medications, several evidences have revealed that this may not necessarily be true [1,2]. Indeed, non-steroidal anti-inflammatory drugs (NSAIDs) aimed at counteracting pain and inflammation in the short term, and antibiotics and anti-asthma, as well as anti-allergic, drugs were prescribed more frequently in elite athletes than in the general population [1,2]. This is a critical point since in several cases there is a misuse of these pharmacological tools without a real need. For this reason, it is important for physicians to educate athletes on the proper use of drugs by illustrating their risks and benefits and informing them about the putative interactions that can occur with other medicines or exercise [1,2].

Moderate-intensity exercise protects individuals from upper respiratory tract infection (URTI), and heavy exertion or prolonged intensive exercise increases the risk of URTI [3], leading athletes to consume oral antibiotics almost twice as often as observed in the general population [1]. This practice is probably aimed at reducing as much as possible the period of inactivity of athletes due to bacterial diseases [2]. 

Antibiotics represent the class of drugs with one of the highest therapeutic indexes, thus indicating their relative safety of use; however, it should be noted that they are not exempt from inducing adverse reactions [2]. Indeed, the use of antimicrobial drugs is related to potential adverse effects, some of which are particularly present in athletes, including the potential lengthening of the QT interval, which can result in almost always fatal *torsades de pointes*, aortic dissection, nerve damage, and glycemic fluctuation as well as tendon injuries [1]. In particular, the Achilles tendon appears to be more susceptible to injury, despite several adverse events also being recorded in the rotator cuff, hands, biceps, thumbs, and other locations [4]. The class of antibiotics mainly associated with these adverse reactions is fluoroquinolones. Clinical evidences have indicated that the median duration of treatment before the onset of tendon injury is 6–8 days, but symptoms can occur as early as 2 h after the first dose and up to 6 months after stopping treatment [4].

In parallel, increasing evidences have pointed out the ability of some classes of antibiotics to induce muscle weakness, pain, and a feeling of fatigue upon resuming physical activity conditions that considerably limit athletic performance [1]. Although this condition could be ascribable to the outcomes of the infection, it is worth noting that antibacterial drugs can negatively influence fundamental aspects of normal musculoskeletal activity, thus exposing athletes to more rapid fatigue [4]. For this reason, tailoring a treatment plan aimed at curbing the athlete’s risk with regard to antibiotic use is paramount for their safety and to maintain adequate athletic performance.

Based on these premises, the aim of this review is to illustrate and critically analyze the molecular mechanisms underlying the onset and development of muscle–tendon alterations in athletes treated with antibacterial drugs, as well as to delineate pharmacological strategies aimed at counteracting such adverse events.

## 2. Antibiotics: Classification, Mechanisms of Action, and Adverse Effects

Antibiotics are considered a cornerstone of modern pharmacology and their discovery has allowed the resolution of infectious diseases [5]. Initially, antibiotics were organic compounds produced by one microorganism in order to counteract the growth of other microorganisms (bacteriostatic) or to kill other bacteria (bactericidal) [6]. Over the years, this definition has been modified to include antimicrobials produced partly or entirely in a synthetic manner [6].

The most common scheme for classifying antibiotics is based on their molecular structure, mode of action, and spectrum of activity [5]. Of note, antibiotics belonging to the same structural class generally display a similar pattern of efficacy and potential side effects [5]. The main classes of antibiotics include β-lactams, aminoglycosides, tetracyclines, fluoroquinolones, macrolides, sulfonamides, and glycopeptides [5] (see Table 1).

## 3. Antibiotics and Altered Mitochondria Physiology in Skeletal Muscle

In recent years, increasing interest has been focused on mitochondria and mitochondrial DNA. In this regard, the endosymbiotic theory of mitochondrial origin postulated that it took place about 1.5 billion years ago and was related to the increase in the O_2_ level in the atmosphere [8]. In particular, such a theory states that mitochondria are ancestors of ancient endosymbiotic organisms and symbiont-resembling bacteria as we know them today [8]. After many evolutionary stages, the mitochondria have passed from being free-living bacteria to becoming an integrated part of the cell as an organelle. Following various endosymbiotic phases, the mitochondria developed transport proteins, cristae structure, biochemical pathways (e.g., glycolytic pathways and lipid synthesis), and division mechanisms that were integrated between symbiont and host cells [8]. At present, mitochondria represent a critical hub for human cells, holding a critical role in energy production through the synthesis of adenosine triphosphate (ATP) in oxidative phosphorylation (OXPHOS). In parallel, a significant amount of ROS, which have a high oxidation potential, is generated [8].

Of note, mitochondria are fascinating organelles regulating many critical cellular processes for skeletal muscle physiology [9]. Indeed, mitochondria play central roles in muscle cell metabolism, energy supply, the regulation of energy-sensitive signaling pathways, ROS production/signaling, calcium homeostasis, and the regulation of apoptosis [9]. It is not surprising that mitochondrial dysfunctions are involved in a large number of adverse events/conditions affecting skeletal muscle health [9].

The repeated, intense use of muscles leads to a decline in performance known as muscle fatigue [7]. In order to characterize the molecular mechanism underlying the decline in performance, the following have been identified: (*a*) the traditional explanation, based on the accumulation of intracellular lactate and hydrogen ions causing the impaired function of the contractile proteins; (*b*) alternative explanations, characterized by ionic alterations affecting the action potential, inducing a failure of sarcoplasmic reticulum Ca^2+^ release by various mechanisms, and the increase in reactive oxygen species (ROS) [7].

Various metabolites are produced as a result of muscle activity, including a large amount of ROS, which plays a critical role in fatigue. The bactericidal activities of the main classes of antibiotics are, at least in part, ascribable to the induction of bacteria death producing detrimental levels of ROS, affecting the tricarboxylic acid (TCA) cycle and electron transport chain (ETC). On this basis, it is conceivable that several classes of antibiotics, acting on the mitochondrial ETC, trigger massive production of ROS in tissues particularly rich in mitochondria, including skeletal muscle, thus triggering fatigue (Figure 1).

*β-lactams.* Several evidences have focused their attention on the detrimental effects exerted by antibiotic therapy on athletic performance [1]. Indeed, several evidences in the scientific literature have reported the ability of some classes of antibiotics to determine muscle weakness and pain, with detrimental effects on athletic activity [1]. Increasing evidences have pointed out that antibiotics target mitochondria and mitochondrial components, similar to their action in bacteria [10]. Indeed, previous studies performed in mammalian systems have revealed parallel antibiotic-target interactions in mitochondria. It has been observed, for instance, that clinically relevant doses of β-lactams, beyond exerting their bactericidal effect inhibiting bacterial cell wall synthesis, are also able to affect mitochondrial carnitine/acylcarnitine transporter activity, as well as induce ROS production in mammalian cells, leading to DNA, protein, and lipid damage [10]. A recent paper by Suzuki et al. demonstrated that transforming growth factor-β (TGF-β)-activated kinase 1 (TAK1) mediated cefotaxime-induced ROS generation [11]. TAK1 is widely known as a stress-responsive kinase that activates the c-Jun N-terminal kinases (JNK) and p38 mitogen-activated protein kinase (MAPK) but not the extracellular-signal-regulated kinase (ERK) pathways [11].

In parallel, it has also been observed that the detrimental effect of β-lactams is based on a dual mechanism of interaction: (*i*) the competitive mechanism determined by the reversible interaction of the pharmacological agents with the substrate binding site of the transporter; and (*ii*) the irreversible mechanism due to the covalent reaction of the compounds with the specific amino acid residues of the protein [12]. The competitive inhibition observed at short times of incubation should be related to structural similarities between the β-lactams and carnitine [12]. It is very likely that in vivo, the irreversible inactivation of the carnitine/acylcarnitine transporter will variably impair the β-oxidation depending on the fraction of inactivated protein [12]. This could lead to consequences on the metabolism of fatty acids for energy production in several tissues, including the liver or muscles, contributing to some mild side effects of β-lactams [12,13].

*Aminoglycosides.* The aminoglycosides are listed by the WHO as critically important antimicrobials for human therapy [14]. In particular, these antibiotics bind irreversibly to the 30S ribosome to interfere with the reading of the microbial genetic code and inhibit protein synthesis [15]. Aminoglycosides are generally bactericidal and their efficacy in several cases can be greatly enhanced by the concomitant use of cell-wall–inhibiting β-lactams and glycopeptides [15].

A pioneering in vitro study by Weinberg et al. showed that gentamicin, neomycin, kanamycin, and streptomycin treatments stimulated State 4 mitochondrial respiration and inhibited State 3 and 2,4-*dinitrophenol* (DNP)-uncoupled respiration, thus providing the first evidence of the detrimental effect of aminoglycosides on ETC [16]. Further investigations revealed that aminoglycosides uncoupled the hydrolytic activity of the catalytic moiety (F1) from the transmembrane proton conduction by the membrane sector (F0) of the ATPase complex [17]. Recently, it has been demonstrated that gentamicin reduced the respiratory control ratio and contextually determined the collapse of the mitochondrial membrane potential [18].

In parallel, it has also been reported that gentamicin induced the activation of the apoptosis markers caspase-3 and -7 [14]. It is worth noting that caspase-dependent and -independent pathways are likely involved in apoptotic nuclear loss during atrophy in skeletal muscle [19]. In particular, caspase-3 activation and subsequent apoptosis occur through the release of cytochrome-*c* from mitochondria in response to changes in the ratio of the pro- and anti-apoptotic members of the Bcl-2 family of proteins [19]. It has been reported that gentamicin is able to alter the cytochrome-*c* complex [18] as well as stimulate the gene expression of the pro-apoptotic marker *Bak* [20]. Of note, it is widely recognized that alterations of the cytochrome-*c* complex are related to a reduction in muscle mass and strength [21,22,23], supporting the evidence of a detrimental effect of aminoglycosides on skeletal muscle physiology.

*Tetracyclines.* Tetracyclines, discovered in the 1940s, exhibited activity against a wide range of microorganisms including the Gram-positive and Gram-negative bacteria, chlamydiae, mycoplasmas, rickettsiae, and protozoan parasites [24]. These antibiotics specifically inhibit the 30S ribosomal subunit, hindering the binding of the aminoacyl-tRNA to the acceptor site on the mRNA–ribosome complex [24].

Several evidences have shown that other antibacterial agents, such as tetracyclines, can affect the physiological activity of skeletal muscle [25]. In particular, the mechanisms of tetracycline action in neural and neuromuscular disorders have typically been believed to be its anti-apoptotic properties, particularly through mitochondria [25].

Subsequently, it has been observed that interleukin (IL)-17 increases the expression and production of metalloproteinases (MMP)-9 in murine myoblast cells, although not significantly affecting the constitutively expressed MMP-2 [26]. It has also been shown that IL-17 increases MMP-9 expression at both the transcriptional and protein levels. It is known that proinflammatory cytokines are able to activate muscle satellite cells, possibly by inducing MMP-9 expression [26]. Doxycycline strongly inhibits IL-17-induced MMP-9 expression and ERK1/2 activation [27]. In addition, it has been reported that the pharmacological inhibition of MEK1,2-ERK1/2 reverted the suppressive effect of IL-17 on myogenic differentiation [27]. Such a pharmacological blockade enhanced myoblast cell differentiation, thus suggesting that doxycycline may rescue myogenic differentiation by downregulating the IL-17 activation of ERK1/2 signaling, critically involved in the increased expression of MMP-9 [27]. With regard to the effects of doxycycline on the murine neuromuscular junction, treatment with this drug does not affect cholinesterase activity or cause damage to skeletal muscle [28]. In particular, doxycycline acted on the ryanodine receptor, sarcolemmal membrane, and neuronal sodium channel with post-junctional consequences due to the decreased availability of muscle nicotinic acetylcholine receptors [28].

Skeletal-muscle-cultured cells treated with minocycline, a tetracycline with a broad spectrum, showed a reduction in both myosin heavy chain content and protein synthesis without visible changes to myotube morphology [25]. In an ex vivo study, the tibial muscle isolated from C57Bl/6 mice administered with minocycline showed no changes in terms of muscle fatigability and force frequency. In contrast, a reduction in the maximum force has been reported [25]. Based on these premises, future studies should be aimed at investigating the efficacy of minocycline treatment in neuromuscular or other muscle-atrophy-inducing conditions.

Recently, it has emerged that tetracyclines can induce mitochondrial proteotoxic stress, leading to changes in nuclear gene expression and altered mitochondrial dynamics and function [29]. Interestingly, the effect of minocycline on mitochondria is related to its ability to chelate Ca^2+^, partially uncouple mitochondria by the formation of ion channels, and, finally, prevent Ca^2+^ accumulation in the mitochondrial matrix [29].

*Fluoroquinolone.* Fluoroquinolone antibiotics are employed to manage a series of infections, such as those of the urinary, respiratory, and gastrointestinal tracts, as well as those of the skin, bones, and joints [30]. Despite the fact that the etiology of fluoroquinolone-associated muscle disorders is yet to be completely clarified, there is evidence of an interplay between latent myopathy disorders and the fluorine atom in fluoroquinolones [30].

Consistent with this view, myalgia represents the most common adverse effect of fluoroquinolone administration [30]. Symptoms, which typically consist of diffuse muscle pain with or without weakness [30] with perhaps a preference for proximal muscle groups [30], appearing within 1 week of fluoroquinolone treatment [30] and often resolving within 1–4 weeks of the discontinuation of the drug [30] despite the persistence of symptoms for up to 6 months, have been described [30]. Of note, the administration of statins can exacerbate fluoroquinolone-associated myopathy [30].

In a case report study, Guy et al. [31] reported that patients with fluoroquinolone-associated myalgia and weakness displayed similar metabolic abnormalities. In particular, norfloxacin elicited severe myalgia and rhabdomyolysis in a patient susceptible to malignant hyperthermia. Myopathic syndrome was associated with drug exposure and an intrinsic tendency to malignant hyperthermia [31]. This type of pharmacogenetic myopathy is often latent with diverse clinical symptoms. There are clinical and physiological clues eliciting a similar muscle deficit, predisposing to malignant hyperthermia and rhabdomyolysis induced by this fluoroquinolone [31].

The possible role of fluorine in this fluoroquinolone-induced rhabdomyolysis is supported by the observation that no muscular adverse effects have been reported during treatment with nonfluorinated quinolones, which were widely used in the past [31]. In muscle physiology, the fluoride ion plays a role in Ca^2+^-dependent K^+^ channels, and probably in calcium release channels (ryanodine receptor)7 that are known to be affected in the skeletal muscles of malignant hyperthermia-susceptible subjects. K^+^ efflux and increased intracellular Ca^2+^ levels are observed in experimental models when fluoride ions interact with calcium receptors [31].

In a set of in vitro experiments, ciprofloxacin affected cellular growth and differentiation in a skeletal muscle cell line [32]. The retardation of cell division was quick and only present in cells with functional mitochondria. This evidence shows a retrograde signal from mitochondria to the nucleus induced by oxidative stress or impaired mtDNA replication [32]. Analogously, the differentiation of myoblasts to multinucleated muscle fibers was impaired upon the inhibition of mitochondria but not nuclear Top 2 [32]. This differentiation process does not require cellular proliferation but is dependent on mitochondrial function, further supporting a signaling connection between the mitochondria and nucleus [32] and thus providing further evidence about the detrimental effect of fluoroquinolones on skeletal muscle.

*Macrolides.* Macrolides, used in the treatment of various bacterial infections, are characterized by a macrocyclic lactone of different ring sizes to which one or more deoxy-sugar or amino sugar residues are attached [33]. The antibacterial activity of macrolides is ascribable to the ability of this antibiotic to bind to the bacterial 50S ribosomal subunit and thus interfere with protein synthesis [33]. The high affinity of macrolides for bacterial ribosomes, together with the highly conserved structure of ribosomes across all of the bacterial species, is consistent with their broad-spectrum activity [33].

Recently, to better understand the targets and effects of this drug class in mammalian cells, Gupta et al. used a genome-wide shRNA screen to identify genes that modulate cellular sensitivity to josamycin, showing that this drug induced impaired oxidative phosphorylation and metabolic shifts to glycolysis [34].

## 4. Antibiotics and Tendinopathy: The Real Achilles Heel?

Tendinopathies represent distinctive and peculiar side effects of fluoroquinolones. Indeed, the administration of these drugs is associated with a significant risk of tendonitis and tendon rupture, with the incidence of tendon injury among those taking fluoroquinolones estimated to be between 0.08 and 0.2% [35]. The pathogenesis of quinolone-induced tendinopathy is considered to be multifactorial [36] and it is usually linked with one or more synergistic factors, such as male sex, age, renal disease, rheumatic disease, co-prescription of corticosteroid, and physical activity [36]. For this reason, some sports medicine specialists have advised the avoidance of fluoroquinolones for athletes [36]. Of note, heterogeneity in the tendon toxicity of fluoroquinolones has been reported, with a greater detrimental effect for ofloxacin and pefloxacin and a reduced risk for levofloxacin [36]. This collagen pathology is emerging as the result of a combination of different factors, including (a) ischemic condition, (b) degradation of the tendon matrix, and (c) toxic alteration of tenocyte activity (Figure 2) [37].

Fluoroquinolone treatment has been shown to dramatically decrease the amount of HIF-1α mRNA. It is worth noting that HIF-1α is a “safety mechanism” involved in switching cell metabolism into the anaerobic pathway in order to protect the cell against oxidative stress [38]. For this reason, it is conceivable that, with this protein not being properly expressed, cells exposed to fluoroquinolones are unable to switch to the anaerobic pathway when necessary, leading to an altered electron transport chain in the mitochondria, which ultimately causes oxidative stress [38]. In addition, it has been demonstrated that Zn^2+^, Cu^1/2+^, Se^2+^, Fe^2/3+^, and Mn^2+^, important cofactors of anti-oxidative enzymes, are chelated by fluoroquinolones, thus exacerbating oxidative damage [38]. In particular, Mn^2+^ chelation has a negative effect on mitochondrial function since Mn^2+^ ensures protection against mitochondria DNA damage by superoxide dismutase 2 (SOD2) [38].

In parallel, fluoroquinolones have displayed anti-angiogenic properties. Indeed, these antibiotics reduced the levels of IL-8 and vascular endothelial growth factor (VEGF) secretion from cells [39,40]. IL-8, stimulating VEGF expression through the activation of ERK and phosphoinositide 3-kinases (PI3K), contributed to angiogenesis and thus to the homeostasis of tendons [41]. For this reason, the detrimental effect of fluoroquinolones on tendons is, at least in part, ascribable to the inhibitory effect of this drug on the IL-8/VEGF axis.

The tendon is a dynamic tissue subjected to continuous remodeling, which is shaped by several factors such as tendon compression, strain level, and shear force [42]. These include both intrinsic factors, including the degeneration of the tendon, and extrinsic influences, for example, micro-trauma from poor body mechanics, athletics, and occupational tendon overuse, which can increase the rate of matrix remodeling and create dysfunction and inappropriate tendon adaptation and repair [42].

A set of experiments from Sendzik et al. demonstrated that the tendons of fluoroquinolone-treated rats exhibited ultrastructural alterations in the extracellular matrix (ECM), including decreased diameters of collagen fibrils and increased distance between the collagen fibrils as well as in the tenocytes [43]. Such morphological findings in rat tendons were confirmed by the results from biochemical studies performed with Achilles tendon samples from ciprofloxacin-treated dogs. In this context, a reduction in the components of the ECM as well as the transmembrane signal receptor of the Map-kinase pathway, β_1_-integrin [44], has been observed. In addition, fluoroquinolones are shown to have a sequestering effect on Mg^2+^, thus affecting the connective structure of tendons [44]. Indeed, Mg^2+^ plays an essential role in a wide range of biological processes and is crucial for connective tissue homeostasis [44]. Further evidences have demonstrated a chelating effect of the fluoroquinolones toward the Fe ion, thus limiting the physiological activity of prolyl 4-hydroxylase (P4HA1) and lysyl hydroxylase (LH1), two iron-dependent enzymes, essential for the post-translational modification of collagen [45]. In particular, transcript analysis of P4HA1 and LH1 showed repression on fluoroquinolone treatment, indicating additional mechanisms involved in collagen weakening [45].

The increase in protease activity and the inhibition of both cell proliferation and the synthesis of matrix ground substances contribute to the clinically described tendinopathies associated with ciprofloxacin therapy [46]. In in vitro experiments performed on canine Achilles tendons, ciprofloxacin inhibited fibroblast matrix synthesis and fibroblast proliferation and increased fibroblast-derived matrix-degrading proteolytic activity [46]. Beyond a detrimental effect of fluoroquinolones on the ECM structures, it has been described that ciprofloxacin decreased the ability of tenocytes for cell–cell or cell–matrix adhesion, downregulating CX43 mRNA levels and thus providing the reduced gap junction communication efficiency and synthetic responsiveness of the ECM [47].

When the tendon matrix is changed, an imbalance and upregulation of MMPs, such as MMP-1, MMP-2, MMP-3, and MMP-13, prompt changes within collagen cross-links and structures [42]. These changes lead to tendinopathy with the potential for tendon rupture. Data from animal studies can be extrapolated and applied to the human model because of similarities in histopathologic reactions, such as dilated blood vessels, edema, the overabundance of mononuclear cells, the rupture of tenocytes and micro-nuclear structures, inflammatory cytokines, and the infiltration of the matrix by ROS [42].

MMP-1 protein levels were increased by ciprofloxacin, whereas no effect was observed on MMP-2 activity. In contrast, a reduction in tissue inhibitor of metalloproteinase (TIMP)-1 mRNA was observed in ciprofloxacin-treated samples. However, another study showed that ciprofloxacin also upregulated the expression of MMP-2 in tendon cells with a concomitant degradation of type I collagen, suggesting a negative effect on tendon structure or its healing process through the mechanism by upregulation of gelatinase (MMP-2) expression [48]. In parallel, ciprofloxacin also potentiated the IL-1–stimulated expression of MMP-3 at both the mRNA and protein level [49].

## 5. Therapeutic Strategies Aimed at Counteracting Tendon and Muscle Alterations Associated with Antibiotic Use

Over the years, increasing evidences have shown that acute and chronic stress induced structural and functional damage to mitochondria, thus redefining the pathophysiological role of such organelles. As previously described, mitochondrial alterations induced by several antibiotics represent a pivotal step in the onset of muscular alterations in athletes. Nowadays, increasing attention has been paid to the development of pharmacological strategies based on *small molecules* active on mitochondria (Figure 1) [50]. Indeed, several studies have focused on the design of novel entities able to selectively act on mitochondria based on the biophysical properties of such organelles, for example, the internal negative potential and the presence of peculiar enzymes or transporters able to bioactivate or internalize prodrugs, respectively, into mitochondria.

Analogously, the aminoglycosides also alter the mitochondrial respiratory chain. For this reason, the use of a series of food supplements capable of supporting this function, including coenzyme Q10 (CoQ10), taurine, vitamin E, and selenium [51], could exert a protective effect. In pre-clinical evidences, coenzyme Q10 administration was found to counteract neomycin toxicity in a vestibular cell line [52]. Similarly, treatment with taurine or selenium showed a protective effect in cells exposed to gentamicin [53,54]. In contrast, conflicting results were observed for supplementation with vitamin E [55].

Macrolide administration, interfering with mitochondrial DNA and proteins associated with mitochondrial DNA, affects oxidative phosphorylation [55]. Some vitamins such as calcitriol could exert a protective effect if administered concomitantly with the antibiotic [55]. A pre-clinical study performed on mitochondria isolated from murine cardiac muscles demonstrated a protective effect of calcitriol on the mitochondrial adverse effects induced by erythromycin [56]. Although no direct evidence is available, the widely recognized role of B vitamins in supporting mitochondrial homeostasis suggests a potential protective use of such a vitamin group during therapy with macrolides [56]. However, further pre-clinical and clinical studies are needed to corroborate this hypothesis.

It is known that mitochondria simultaneously represent one of the main sources and targets of oxidative stress. Inadequate antioxidant protection could lead to mitochondrial dysfunction, which can contribute to fluoroquinolone-induced tendinopathy [57]. For this reason, growing interest from the scientific community is turning to the identification of novel therapeutic strategies able to prevent fluoroquinolone-related tendon damage without affecting the antibacterial effect [57]. In this regard, it has been observed that CoQ10 is the predominant form of the electron and is a ubiquinone carrier, playing an important role in the protection of mitochondria from oxidative stress [57]. Based on these premises, pharmacological research developed MitoQ, a deci-lubiquinone covalently linked to the TPP (triphenylphosphonium) cation, which is able to selectively target the mitochondrion with a putative protective effect on the tendon during pharmacological therapy with fluoroquinolones [57]. A recent study evaluated both the ability of moxifloxacin and ciprofloxacin in mitochondrial dysfunction in human tendon cell cultures and the protective effect of MitoQ in comparison with idebenone, a non-mitochondrial form of ubiquinone [57]. This study confirmed that both ciprofloxacin and moxifloxacin caused oxidative stress and damage to the mitochondrial membrane contributing to the development of tendonitis and tendon rupture in some patients [57]. Furthermore, it emerged that MitoQ exerted a protective effect in this context, thus suggesting the use of MitoQ as a protective strategy to antagonize fluoroquinolone-induced damage [57].

## 6. Discussion and Conclusions

In analyzing the prodromal molecular basis of muscle–tendon alterations associated with the use of antibiotics, it emerged that there are two molecular mechanisms: (1) the production of oxygen radicals (ROS) and (2) mitochondrial dysfunction.

With regard to the production of ROS, it is useful to point out that this mechanism is used by some classes of antibiotics (i.e., fluoroquinolones) to carry out their antimicrobial action. However, when this product exceeds the detoxifying capacity of the tissue, such biochemical alterations can induce marked damage to the muscle–tendon compartment.

Over the years, the identification of antibiotics as the cause of ROS overproduction and mitochondrial dysfunction in mammalian cells has provided a basis for the development of novel therapeutic strategies that are useful for alleviating the adverse side effects associated with antibiotics. It is conceivable that since an accumulation of ROS represents one of the main conditions capable of eliciting the aforementioned alterations, administration with antioxidant supplements could represent a valid alternative. However, a significant intake of these compounds could affect the antimicrobial capacity of some classes of antibiotics that use ROS production as a bactericidal mechanism.

For this reason, there is a growing interest in the scientific community in the development of pharmacological strategies based on standard molecules or *small molecules* that selectively act on mitochondria. In this regard, various pre-clinical evidences have shown that supplementation with molecules supporting the fundamental molecular mechanisms for mitochondrial respiration (CoQ10, taurine, vitamin E, MITO-Q, calcitriol) is able to support an adequate transport of carnitine (niacin) and ensure a protective effect on muscle fiber and tendon tissue without a significant alteration of the antimicrobial capacity of antibiotics.

In conclusion, the available evidence derived from pre-clinical studies requires further clinical investigations. However, it emerged that some classes of antibiotics are endowed with a greater propensity to muscle fatigue and tendon injuries and that adequate supplementation with pharmacological tools aimed at supporting mitochondrial homeostasis may be an adequate strategy to mitigate the adverse events in the muscle–tendon compartment.

## Figures and Tables

**Figure 1 sports-10-00131-f001:**
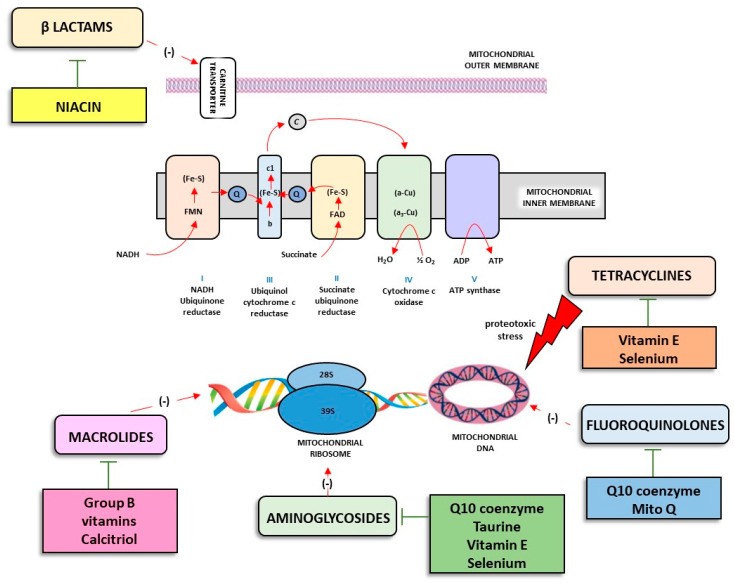
Schematic representation regarding the main steps in mitochondrial activity. In particular, mitochondrial respiration is the set of metabolic reactions and processes requiring oxygen that takes place in mitochondria to convert the energy stored in macronutrients to adenosine triphosphate (ATP). In this context, several antibiotics exert detrimental effects on mitochondrial activity, which can be counteracted by the concomitant administration of several food supplements aimed at restoring mitochondrial homeostasis. Abbreviations: FAD: flavin adenine dinucleotide; FMN: *flavin* mononucleotide; NADH: nicotinamide adenine dinucleotide.

**Figure 2 sports-10-00131-f002:**
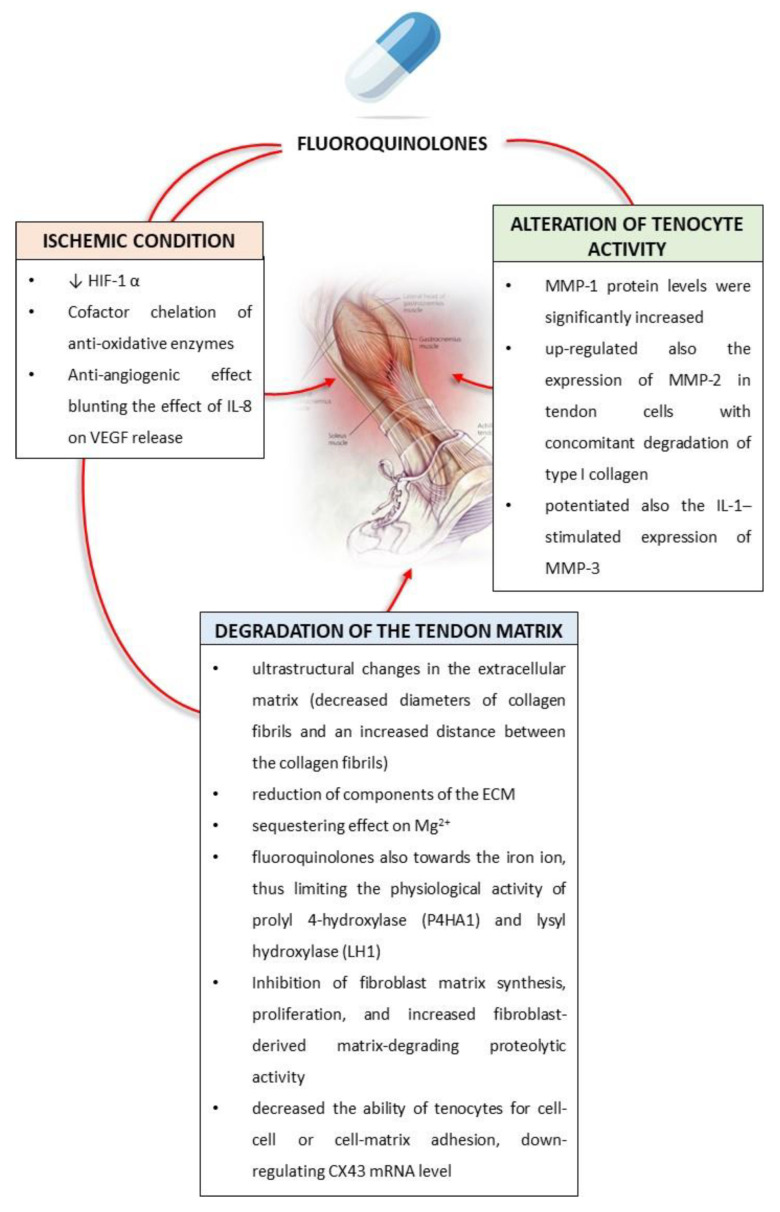
Schematic representation of the main detrimental mechanisms underlying tendon injury induced by fluoroquinolones. Abbreviations: CX43: connexin 43; MMP: metalloproteinase; VEGF: vascular endothelial growth factor.

**Table 1 sports-10-00131-t001:** Pharmacologic characteristics of the main classes of antibiotics.

Drugs	Mechanisms of Action	Resistance Mechanisms	Adverse Events	Ref.
*β lactams*	Bactericidal. Inhibition of bacteria cell wall synthesis interacting with penicillin-binding protein (PBP).	- β lactamases - modification of PBP - efflux pumps - membrane impermeability	Diarrhea, nausea, and vomiting Abdominal pain Allergic reactions Hypersensitivity	[1]
*Aminoglycosides*	Bactericidal. Binding with 30S ribosomal subunit. Reduced translation. Inhibition of protein synthesis.	- modification of antibiotic structure (acylation, phosphorylation, adenylation)	Headache Paresthesia Fever Superinfections Vertigo Skin rash Dizziness	[2]
*Tetracyclines*	Bacteriostatic. Binding with 30S ribosomal subunit. Reduced translation. Inhibition of protein synthesis.	- efflux pumps - ribosomal protection - enzymatic inactivation	Discoloration of teeth Enamel hypoplasia Diarrhea Nausea Photosensitivity Stomach upset Loss of appetite White patches or sores inside mouth or on lips Swollen tongue	[3]
*Fluoroquinolones*	Bactericidal. Targets of GyrA subunit of DNA gyrase and topoisomerase IV. Inhibition of DNA synthesis	- efflux pumps - target modification	Digestive disorders Hyperglycemia Hypoglycemia QTc prolongation and cardiac arrhythmia Retinal detachment Tendinopathy Tendon rupture Peripheral neuropathy Aortic aneurysm	[4]
*Macrolides*	Bacteriostatic. Binding with 50S ribosomal subunit. Inhibition of protein synthesis.	- target modification - enzymatic inactivation (phosphotransferases, glycotransferases, esterases)	Allergic reactions Cholestatic hepatitis Cardiac arrhythmias Transient auditory impairment	[5]
*Carbapenems*	Bactericidal. Atypical β-lactam antibiotics with broad-spectrum high antibacterial activity.	extended spectrum β-lactamases	Nausea and vomiting Seizures Patients with allergies to other β-lactams may experience hypersensitivity reactions	[6]
*Trimethoprim*	Bacteriostatic. Inhibition of folate synthesis. Dihydrofolate reductase (DHFR) inhibition.	- a resistant form of DHFR - mutations of gene promoter and increase of upstream signals codifying for DHFR	Itching and rash Diarrhea Nausea Vomiting Stomach upset Loss of appetite Changes in taste Headache Skin sensitivity to sunlight Swollen tongue Fever	[7]

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
