# Peer review of "Antibiotic Therapy and Athletes: Is the Mitochondrial Dysfunction the Real Achilles’ Heel?"

_sports, 2022, doi:10.3390/sports10090131_

Round 1

Reviewer 1 Report

1.  English sentence structure too wordy and complicated. Examples:

A. Starting numerous sentences and paragraphs with "despite" (lines 22,38,32,45,70) complicates the structure

B. Grammar: line 115

C. Unnecessary phrases at sentence starts: i.e. 117-118, "it is worth to note" (line 288)

D. Redundant phrases: "in particular" and "for instance" (line 131); "similarly to other" and "also" (line 181)

E. Confusing how written: i.e. lines 282-285, 316-319

F. 6/8 --> 6-8 (line 48)

2. Spell out all abbreviations - JNK, MAPK, ERK (line 138); SOD2 (line 312); PI3K (line 315); TIMP-1 (line 359)

3. Would suggest retitling section 3 to something related to antibiotics and mitochondria as this seems the direct topic of the section

4. Would suggest moving up section 3 paragraphs 3 & 4 that discuss overview of mitochondria to start of section

5. Consider dividing antibiotic groups with sub-section headings for ease of reading

6. Lines 185-193 may not be necessary to discussion in already lengthy section

7. Section 4: focus more concisely and clearly on the topic of mitochondrial dysfunction in the pathogenesis of tendinopathies. Focus less on other pathogenesis theories. (Figure 2 adequately summarizes others and easier to read to avoid narrative of those)

8. Section 5: Too much recap of intro (i.e. paragraph 2). Subsequent paragraphs are better. VERY interesting section!

9. Section 6: First three paragraphs (except last sentence of 3rd) is a recap of intro, not necessary at this length

10. Section 6: paragraph 4 regarding microbiota is not directly in line with topic of this paper (could be separate paper, interesting)

11. Throughout paper, there are a couple sentences on areas of further research - these statements fit better as part of discussion

Author Response

  1. English sentence structure too wordy and complicated. Examples:
  2. Starting numerous sentences and paragraphs with "despite" (lines 22,38,32,45,70) complicates the structure

As kindly suggested by the reviewer we erased “despite” in several paragraphs

  1. Grammar: line 115

In line with this suggestion, we revised the grammar

  1. Unnecessary phrases at sentence starts: i.e. 117-118, "it is worth to note" (line 288)

As kindly indicated by the reviewer we erased the unnecessary sentences

  1. Redundant phrases: "in particular" and "for instance" (line 131); "similarly to other" and "also" (line 181)

As kindly suggested by the reviewer we erased "in particular" and "for instance" "similarly to other" and "also" in several paragraphs

  1. Confusing how written: i.e. lines 282-285, 316-319

We rephrased the above mentioned lines

  1. 6/8 --> 6-8 (line 48)

We have changed 6/8 into 6-8

  1. Spell out all abbreviations - JNK, MAPK, ERK (line 138); SOD2 (line 312); PI3K (line 315); TIMP-1 (line 359)

As suggested we spelled out all abbreviations

  1. Would suggest retitling section 3 to something related to antibiotics and mitochondria as this seems the direct topic of the section

As kindly suggested we retitled the section 3

  1. Would suggest moving up section 3 paragraphs 3 & 4 that discuss overview of mitochondria to start of section

As indicated we moved up section 3 paragraphs 3 & 4 that discuss overview of mitochondria to start of section

  1. Consider dividing antibiotic groups with sub-section headings for ease of reading

As suggested we divided antibiotic groups with sub-section headings for ease of reading

  1. Lines 185-193 may not be necessary to discussion in already lengthy section
  2. Section 4: focus more concisely and clearly on the topic of mitochondrial dysfunction in the pathogenesis of tendinopathies. Focus less on other pathogenesis theories. (Figure 2 adequately summarizes others and easier to read to avoid narrative of those)

We refocused the section 4 clearly on the topic of mitochondrial dysfunction in the pathogenesis of tendinopathies

  1. Section 5: Too much recap of intro (i.e. paragraph 2). Subsequent paragraphs are better. VERY interesting section!

As suggested we revised the section 5 accordingly with reviewer comment

  1. Section 6: First three paragraphs (except last sentence of 3rd) is a recap of intro, not necessary at this length

As indicated we erased first three paragraphs of section 6

  1. Section 6: paragraph 4 regarding microbiota is not directly in line with topic of this paper (could be separate paper, interesting)

We agree with this suggestion and we erased the section regarding the microbiota

  1. Throughout paper, there are a couple sentences on areas of further research - these statements fit better as part of discussion

We moved the sentences regarding the area of further research into the discussion

Reviewer 2 Report

The proposed manuscript is a well-prepared, comprehensive and concise review, that aims to analyze the effects of antibiotic therapy in athletic population. The review also identifies new strategies in order to counteract potential adverse effects.

The introduction reveals what is already known about this topic and the authors clearly explain the starting background, including appropriate studies.

Successive sections are explained in a very detailed and comprehensive way, providing the reader with adequate information.

Figures are relevant and clearly presented and cited sources are referenced correctly.

I have only one suggestion, maybe the authors could review the formatting of the tables, choosing one that best suits the text.

Author Response

We’d like to thank the Reviewer for his appreciation. We have formatted the tables as indicated.

Round 2

Reviewer 1 Report

Line 32: Delete "A"

Line 38: Delete "The"

Line 206: Delete "other antibacterial agents"

Otherwise, changes make an easier read, thank you